# Urine resazurin reduction ratio as a biomarker of urinary tract infection in people with neurogenic bladder: A first in human study

Abigail P. Fox[1,2]*, Ana Valeria Aguirre Guemez[1,3,4], Christopher Skipwith[5], Courtney Cavin[5], Suzanne L. Groah[3,4,5]

1 MedStar Health Research Institute, Washington, DC, Columbia, United States Of America,
2 Georgetown University School of Medicine, Washington, DC, Columbia, United States Of America,
3 MedStar National Rehabilitation Hospital, Washington, DC, Columbia, United States Of America,
4 MedStar Georgetown University Hospital, Washington, DC, Columbia, United States Of America,
5 Astek Diagnostics Inc., Baltimore, Maryland, United States Of America

* apf61@georgetown.edu

## Abstract

Among people with neurogenic lower urinary tract dysfunction (NLUTD), urinary tract infection (UTI) is the most common infection and a leading cause of morbidity. UTI diagnosis is based on urinary symptoms, bladder inflammation (urinalysis (UA)), and bacterial growth (standard urine culture (SUC)), the latter two assessments taking approximately 24h and 48-72h, respectively. Here, urine resazurin reduction ratio (uRRR), which is a measure of nicotinamide adenine dinucleotide hydrogen conversion, a bioproduct of bacterial metabolism, is assessed as a point-of-care biomarker of UTI with time to result (TTR) of 1 hour. The objectives of this first-in-human study are to evaluate the relationship between the uRRR and: urinary symptoms, urine white blood cell (uWBC) count, urine nitrite, bacterial growth on standard urine culture (SUC), and the likelihood of UTI based on combinations of these. This is a cross-sectional study of participants who completed the Urinary Symptom Questionnaire for people with Neurogenic Bladder (USQNB) and provided urine samples (N=289). Accuracy of uRRR when compared to colony forming units per milliliter (CFU/mL) obtained by SUC was 92.73%. A significant relationship was found between higher uRRR and positive uWBC (p<0.0001), positive nitrite (p<0.0001), and symptoms (p<0.0001). uRRR values were significantly higher in the UTI positive group, which was defined as a combination of positive symptoms, bladder inflammation, and bacterial growth. In this first report of uRRR as a potential biomarker and rapid diagnostic assessment of UTI among people with NLUTD, uRRR was associated with each of the standard of care diagnostic assessments individually and when combined into UTI likelihood groups. Given these encouraging preliminary data, next steps will be to assess uRRR in a prospective clinical trial to determine whether point-of-care uRRR provides clinically actionable information equivalent or superior to standard of care assessments when diagnosing and treating suspected UTI in this population.

**Data availability statement:** All relevant data are within the manuscript and its Supporting Information files.

**Funding:** This work was funded by Astek Diagnostics, Inc., (https://astekdx.com). Astek Diagnostics, Inc. did not play a role in study design or data collection. Astek Diagnostics, Inc. was involved in the analyses, decision to publish and preparation of the manuscript.

**Competing interests:** I have read the journal's policy and the authors of this manuscript have the following competing interests: CS serves as CSO and receives salary from Astekdx; CC serves as COO and receives salary from Astekdx; SLG serves as CMS and has future equity shares in Astekdx. This does not alter our adherence to PLOS ONE policies on sharing data and materials.

## Introduction

Among people with neurogenic lower urinary tract dysfunction (NLUTD), urinary tract infection (UTI) is the most common infection [1] and a leading cause of morbidity [2], emergency room visits, and re-hospitalizations [3]. By definition, UTI in this population is considered a complicated UTI (cUTI) due to the underlying neurologic impairment and the frequent need for catheterization. UTI diagnosis is traditionally based on the presence of urinary symptoms, urinalysis (UA) findings of bladder inflammation, and bacterial growth determined by standard urine culture (SUC) [4]. However, each of these assessments has limitations in cUTI related to NLUTD, which undermines the diagnostic framework [5].

Urinary symptoms are typically altered or subtle among people with NLUTD due to the motor and sensory impairments related to the underlying neurologic diagnosis. There has been significant recent progress in identifying and stratifying which symptoms and symptom complexes are more indicative of UTI [6,7] and therefore under which circumstances laboratory assessment is warranted. However, the standard of care assessments for UTI, UA and SUC, offer limited clinically actionable results for diagnosis of complicated UTI [4,8]. The presence of persistent bladder inflammation and bacterial growth in the absence of symptoms among people with NLUTD renders these assessments poorly informative for UTI diagnosis, except in select cases in which either value is negative [9,10]. As such, the Infectious Diseases Society of America (IDSA) recommends against the use of urine white blood count (uWBC) in differentiating asymptomatic bacteriuria from UTI in this population [4]. Urine culture, while often not warranted for uncomplicated UTI due to high *E.coli* prevalence among these cases, is commonly used in cases of recurrent or complicated UTI (such as with NLUTD). However, the clinical value of SUC has been called into question as comparisons with expanded quantitative urine culture and PCR indicate that it misses 90% of bacteria and 50% of uropathogens [6]. Moreover, time to result (TTR) for both UA and SUC is up to 24 hours and at least 48–72 hours (or longer), respectively, and, as such, clinically actionable results are not available at point-of-care. Lastly, diagnostic tools must align with the new understanding of the urobiome by assessing bacterial activity across the entire urinary microbiota, as opposed to one or two isolated bacterial strains, as with SUC. These current diagnostic limitations underscore the need for more rapid and sensitive diagnostic tools for UTI detection in NLUTD.

As a result of these diagnostic limitations, treatment of UTIs in people with NLUTD is highly subjective [11] and empiric antibiotic use is common, contributing to overtreatment [12]. The latter has not improved UTI prevalence, and has had the unwanted consequence of multidrug-resistant infections, frequently requiring subsequent re-treatment with broad-spectrum antibiotics [13]. There is a need for improved UTI diagnostics that provide clinically actionable information at point-of-care to guide clinical decision making.

This first-in-human study reports on a novel approach to rapid UTI diagnostics that is based on quantification of bacterial metabolic activity as a biomarker of likelihood of UTI. To do so, the oxidation of NADH to NAD$^+$ is measured, which drives the reduction of resazurin to the fluorescent compound resorufin, enabling quantification

via fluorescence spectroscopy [14]and reported in relative fluorescent units (RFU) [15,16]. The objectives of this first-in-human study were to evaluate the relationships between the urine resazurin reduction ratio in urine samples (uRRR) and individual and combined components of UTI diagnosis: urinary symptoms, uWBC count, urine nitrite, bacterial growth on SUC, and the likelihood of UTI based on combinations of these factors.

## Methods

### Preclinical experimental and statistical methods

**Filter-sterilized urine samples.** Filter-sterilized urine samples were used to determine the Limit of Blank (LoB). Filter-sterilized urine samples had low concentrations of four common strains of bacteria added to them to determine the Limit of Quantitation (LoQ) and the Limit of Detection (LoD) (Supplemental Table 1). To prepare filter-sterilized-urine, urine (samples acquired from the University of Maryland, Baltimore Microbiology Laboratory) was drawn up into a 10-mL plastic syringe (BD, Franklin Lakes, NJ). The syringe was attached to a Whatman™ Pop-Top syringe filter adapter (Cytiva Life Sciences, Marlborough, MA) fitted with a PrimeCare™ Separation Membrane (Fortis Life Sciences, Boston, MA), and the urine was pushed through the membrane in a smooth, controlled motion, while collecting the filtrate into a fresh tube. All filter-sterilized urine samples used in experiments were confirmed culture-negative by SUC. Assay linearity was measured using filter-sterilized urine samples spiked with high and low concentrations of four common bacterial strains and the linear regression model confirmed the assay's proportional response across a clinically relevant range of bacterial concentrations ($10^3$–$10^5$ CFU/mL), which validates the assay's analytical sensitivity (Supplemental Fig 1 and Supplemental Table 2 in S1 File). Calibration curves were created using the spiked urine samples (0–$10^6$ CFU/mL of the same 4 bacterial strains) to validate the assay's calibration ($R^2 > 0.95$ within the limits of detection). See Supplemental Methods: Limits of blanks and Limits of detection for details.

**Bacterial culture preparation.** Gram-negative (*E. coli* ATCC 25922 and *K. pneumoniae* ATCC 35657) and Gram-positive (*S. saprophyticus* ATCC 15305 and *E. faecalis* ATCC 29212) reference strains (ATCC; Manassas, VA) were plated on SUC and an isolated colony was grown into exponential phase in tryptic soy broth (TSB). The bacteria were then counted via plating on tryptic soy agar (TSA) and stocks were aliquoted and frozen with 20% glycerol (v/v) at −80 °C. Prior to each experimental run, a fresh aliquot of bacteria was thawed and washed twice with Mueller-Hinton II broth (MHB; Becton, Millipore, St. Louis, MO). The bacterial inoculum was prepared by inoculating a flask containing MHB and shaking overnight at 100 rpm in a shaker incubator set to 37°C. On the day of the experiment, 100 µL of overnight growth was inoculated into a fresh flask containing MHB and placed into the shaker incubator. Absorbance at 600 nm was measured until the culture reached exponential growth phase and the bacterial suspension was then diluted with human sterile-filtered urine to $10^7$ CFU/mL. Ten-fold serial dilutions of this bacterial stock were then prepared with sterile urine. $10^5$ CFU/mL of spiked urine sample was diluted at a 1:1 ratio with MHB and pre-incubated at 37°C for 1 hour. The bacterial samples were subsequently prepared for uRRR analysis.

**uRRR quantification.** The microfluidic system relies on a fluorescence detection assay that uses the capacity of viable bacteria to oxidize NADH to NAD+, thereby reducing non-fluorescent resazurin (alamarBlue) to its highly fluorescent product, resorufin. The conversion rate is proportional to the bacterial metabolic rate and cell density. The highly fluorescent resorufin was monitored using a kinetics fluorimeter in the microfluidic system. uRRR was determined for each sample by measuring the fluorescence signal from total conversion over 15 minutes, normalized by the signal from the background matrix over the same time frame.

**Statistical comparison of uRRR with bacterial growth and symptoms.** A 3x3 table was used to determine uRRR, which was then collapsed into a 2x2 table ($<10^3$ versus $>10^3$) to determine sensitivity and specificity. AUC, sensitivity, and specificity are valid for the evaluation of a marker. Index of Union (IU) was used for threshold discrimination and the defined threshold was then used to calculate sensitivity, specificity, PPV, and NPV for uRRR versus SUC detection for clinical samples (N = 289). Symptom data distributions were described, bivariate associations with outcomes were

examined, and the OC was examined using likelihood ratios, sensitivity, specificity, PPV, and NPV, along with their 95% CI (**Supplemental Methods: Statistical comparison of uRRR and bacterial growth by SUC (using bacterial counts)**. See **Supplemental Material 1 in** S1 File for preclinical experimental and statistical methods continued.

## Clinical methods

This is a cross-sectional study of 289 samples from participants with NLUTD in which symptoms assessment and urine sampling for inflammation, bacterial growth and uRRR was done simultaneously. Approval was obtained from the MedStar Health Research Institute Institutional Review Board (IRB# 00006667). Recruitment took place from June 14th, 2023 to October 30th, 2024 and all participants provided written informed consent before participating. Inclusion criteria are: 18 years of age or older; and NLUTD, managed by primarily indwelling catheterization (IDC; $n = 55$), intermittent catheterization (IC; $n = 108$), or voiding (V; $n = 126$), in both inpatient and outpatient settings. Exclusion criteria included current antibiotic use or antibiotic use in the past 7 days. Participants could provide up to 5 urine samples, with at least one week between samples. Medical history, bladder management type and duration, UTI history, recent antibiotic use, and urine-related symptoms (using the Urinary Symptom Questionnaire for People with Neurogenic Bladder (USQNB)) [6,7] were collected at the time of study enrollment.

Participation in the study involved urine sample collection and completion of the USQNB. Urine samples were collected according to group: by clean catch midstream voided sampling in the V group; directly from a new, unused catheter in the IC group; and from the collection bag in the IDC group. For this first-in-human cross-sectional study, the investigators intentionally deviated from clinical standards and collected urine from the collection bag of those participants who use indwelling catheters. This was done to ensure a wide spectrum of microbial conditions and a sufficient number of poly-microbial samples to rigorously assess device performance under these conditions. This study was intended as an initial assessment of correlations (and not diagnosing UTI) to determine appropriateness for a next-stage human trial. Urine samples were assessed using uRRR, UA, and SUC. Urinalysis and standard urine culture was performed using standard procedures and defined as positive or negative based on the bacterial thresholds listed below. For the purpose of this study, the following definitions were utilized:

- uWBC+ and uWBC- (determined by UA) were ≥6 hpf uWBC and <6 hpf uWBC, respectively;

- CFU+ and CFU- (determined by SUC) were ≥ $1x10^5$ CFU/mL and < $1x10^5$ CFU/mL, respectively;

- Nitrite were reported directly from UA as NIT+ and NIT-.

**Symptom risk profiles.**  Symptoms, symptom complexes, and symptom-based risk profiles were based on the USQNBs(6,7) (V, IC, and IDC versions, respectively). Using the USQNBs, symptoms are grouped into the following categories: action needed (A), bladder (B1), urine quality (B2), and constitutional/other (C) symptoms. These symptom categories are the same across all three USQNBs, and symptoms are exchangeable within but not across categories. In an international multi-phase qualitative study (semi-structured interviews, focus groups, and delphi), symptom-based risk profiles of UTI likelihood (each with the same definition across USQNB) were developed [17]. The highest likelihood symptom profile for UTI risk is defined by the presence of at least one A symptom, at least one B1 or B2, and any number of C symptoms. The lowest likelihood symptom profile is defined as zero symptoms from any category (A, B1, B2 or C). See **Table 1** for all symptom profile definitions.

**UTI risk using symptoms, uWBC and CFU.**  For the purpose of this study, UTI likelihood was defined to be:

- UTI positive (UTI+) = higher or moderate likelihood of UTI according to USQNB-based risk profile, uWBC+, and CFU+;

- UTI Indeterminate = 2 out of 3 of the following: higher or moderate likelihood of UTI per symptoms on the USQNB, uWBC+ and/or CFU+

**Table 1. USQNB-based profiles reflecting combinations of A, B1, and B2 type symptoms.**

| Symptom Type | | | | Profile |
|---|---|---|---|---|
| Actionable (A) | Bladder (B1) | Urine (B2) | Other (C) | |
| At least one | At least one (B1 or B2) | | Any | *Higher likelihood* of symptomatic UTI |
| At least one | Zero | Zero | Any | *Moderate likelihood* of symptomatic UTI |
| **OR** | | | | |
| Zero | At least one | At least one | Any | |
| Zero | At least 4 | Zero | Any | *Low to Moderate likelihood* of symptomatic UTI |
| Zero | Zero | At least one | At least one | *Lower likelihood* of symptomatic UTI |
| **OR** | | | | |
| Zero | 1-3 | Zero | Zero | |
| Zero | Zero | | At least one | *Low likelihood* of symptomatic UTI |
| Zero | Zero | | Zero | *Lowest likelihood* of symptomatic UTI |

- UTI Unlikely = 1 out of 3 of the following: higher or moderate likelihood of UTI per symptoms on the USQNB, uWBC+ or CFU+

- UTI Negative (UTI-) = low-moderate or lower or lowest likelihood of UTI per symptoms on the USQNB, uWBC-, and CFU- (**Table 2**).

 **Data analysis and plotting.** Data was loaded into a SQL database to be filtered and analyzed. Plots and statistical testing were performed using GraphPad Prism version 10.4.0 for MacOS (GraphPad Software, www.graphpad.com) and SPSS Statistics version 29.02 for MacOS (IBM, www.ibm.com/spss). Data was loaded into the R statistical environment for analysis of identified bacterial strains. Heat maps were generated using the ggplot2 package. Figs were finalized in Adobe Illustrator.

## Results

### Preclinical evaluation of urine resazurin reduction ratio (uRRR) quantification

The urine resazurin reduction ratio (uRRR) was calibrated using Gram-negative and Gram-positive reference strains spiked into human filter-sterilized urine. The concentrations of each strain were varied to calibrate a dose-response. The dose-dependent direct correlation between RFUs and bacterial density was observed in these bacteria-spiked samples

**Table 2. UTI definitions based on symptom profile, inflammation and bacterial growth.**

| Symptom Profile | UTI+ | UTI- | UTI Indeterminate | UTI Unlikely |
|---|---|---|---|---|
| **Symptom+** | | | 2 out of 3 of: | 1 out of 3 of: |
| *High* | x | | • *higher or moderate symptoms* | • *higher or moderate symptoms* |
| *Moderate* | x | | • *WBC+* | • *WBC+, or* |
| **Symptom-** | | | • *CFU+* | • *CFU+* |
| *Low-moderate* | | x | | |
| *Lower* | | x | | |
| *Low* | | x | | |
| *Lowest* | | x | | |
| **WBC** | pos | neg | | |
| **CFU** | pos | neg | | |

(Fig 1A). Assay linearity for uRRR measurements was demonstrated by mixing a high concentration spiked urine sample ($10^5$ CFU/mL) with a low concentration spiked urine sample ($10^3$ CFU/mL) at varying proportions to assess the proposed linear range. Linearity measurements demonstrate the relationship between signal and bacterial concentration in human sterile-filtered urine samples containing different levels of bacteria (Fig 1B). The uRRR exhibited a linear correlation in the range of $10^3$ to $10^5$ CFU/mL for urine spiked with *Escherichia coli* ($R^2 = 0.9575$, $p < 0.0001$), *Staphylococcus saprophyticus* ($R^2 = 0.9154$, $p < 0.0001$), *Klebsiella pneumoniae* ($R^2 = 0.9548$, $p < 0.0001$), and *Enterococcus faecalis* ($R^2 = 0.9195$, $p < 0.0001$) (Fig 1B). See **Supplemental Results: Evaluation of uRRR quantification in** S1 File.

## Preclinical determination of uRRR cut points

uRRR fluorescence values corresponding to calibration values were compared to SUC-determined CFU/mL values for single-species clinical samples ($n = 199$). Data was reported in a 3x3 contingency table of frequencies and percentages. The three categories for both methods (uRRR versus SUC) were $\leq 10^3$; $> 10^3$ and $< 10^5$, and $\geq 10^5$. Agreement between the two methods was assessed using a quadratically weighted Cohen's κ statistic. The calculation of weighted κ assumes the categories are ordered and accounts for how far apart the two raters are. The weighted Cohen's κ statistic of 0.758 (95% confidence interval, 0.638 to 0.803) indicated substantial agreement between uRRR-calibrated reference ranges and those from SUC. Accordingly, uRRR rates utilized in the clinical study below were defined as: positive $\geq 0.7310$ RFU (corresponding to $\geq 10^5$ CFU/mL), intermediate $> 0.5469$ and $< 0.7310$ RFU ($> 10^3$ and $< 10^5$ CFU/mL), and negative $\leq 0.5469$ RFU ($\leq 10^3$ CFU/mL).

## Demographics of clinical cohort

Participants (N = 213) provided N = 289 urine samples, stratified by bladder management: IDC (n = 55), IC (n = 108), V (n = 126) (Table 3). Table 3 summarizes the participant population.

## Relationship between uRRR and bacterial growth on SUC

Overall accuracy of uRRR when compared to CFU/mL obtained by SUC was 92.73%. The sensitivity was 86.44%, specificity 97.08%, positive predictive value 95.33%, and negative predictive value 91.21% (Table 4). The agreement between RFU values and CFU/mL was assessed using a quadratically weighted Cohen's κ statistic. The weighted Cohen's κ statistic of 0.758 (95% confidence interval, 0.638 to 0.803) indicated substantial agreement between uRRR-calibrated reference ranges and those from SUC. A Kolmogorov-Smirnov test showed a significant difference in uRRR values between CFU- and CFU+ groups ($D = 0.9605$, $p < 0.0001$) (Fig 2A). The true positive rate (TPR) was compared to the false positive rate (FPR) of uRRR. The area under the curve (AUC) of the receiver-operating characteristic (ROC) analysis was 0.9785 (with the 95% confidence interval 0.9579–0.9990, $p < 0.0001$) (Fig 2B).

## Relationship between uRRR results versus individual biomarkers (i.e., nitrite and uWBC count)

Kolmogorov-Smirnov tests showed significant differences in uRRR values for NIT+ versus NIT- ($p < 0.0001$) (Fig 2C) and uWBC+ versus uWBC- ($p < 0.0001$) in participant samples (Fig 2D).

## Relationship between uRRR and symptom profiles

Low and high symptom profiles stratified by bladder management cohort (V, IC, and IDC) were compared to uRRR. USQNB profiles of low and lowest likelihood of symptomatic UTI were combined to constitute the low risk (L) group, while the USQNB profile of highest likelihood of symptomatic UTI constituted the high risk (H) group (see Table 1). Across all three bladder management groups, Kolmogorov-Smirnov tests showed significantly higher uRRR values in the "high/moderate-high likelihood of UTI" (H) group than the combined "low likelihood of UTI" (L) group in each bladder management cohort (V, $p < 0.0001$; IC, $p = 0.0004$, IDC, $p < 0.0001$) (Fig 3) (**Supplemental Methods: Statistical comparison of uRRR and symptom categories** in S1 File).

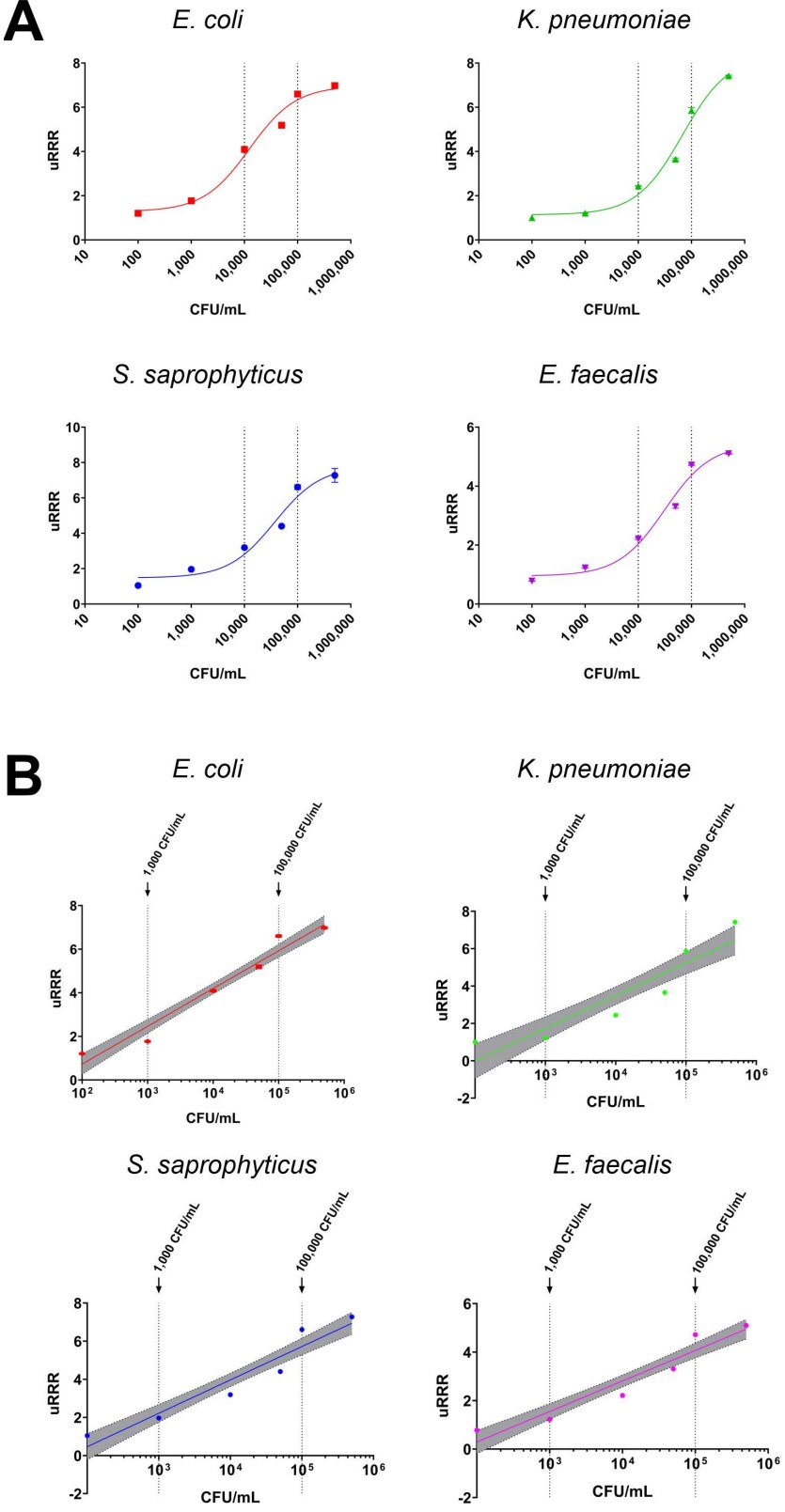

**Fig 1. Analytical sensitivity of uRRR with uropathogenic bacteria. (A)** Dose-response curves of sterile-filtered urine spiked with *Escherichia coli (red)*, *Klebsiella pneumoniae* (green), *Staphylococcus saprophyticus* (blue), or *Enterococcus faecalis* (magenta). **(B)** Relative fluorescence units (RFU)

measure total resazurin reduction over 15 minutes, normalized by the signal from the background matrix to derive uRRR. Shown are the mean of four measurements with standard deviation error bars, and line of best fit generated using linear regression with 95% confidence interval.

**Table 3. Participant demographics.**

| Age (mean ± std dev) | 53.22 ± 16.11 |
|---|---|
| Female | 91 |
| Male | 198 |
| **Bladder Management** | |
| Indwelling Catheter | 55 |
| Intermittent Catheter | 108 |
| Void | 126 |
| **Diagnosis** | |
| Spinal cord injury/disease | 266 |
| Multiple Sclerosis | 18 |
| Spina Bifida | 5 |

**Table 4. Sensitivity, specificity, PPV, and NPV of uRRR versus SUC by bladder management group.**

| | IDC | IC | V | Total | Percent |
|---|---|---|---|---|---|
| *Concordance* | | | | | |
| True Positive | 28 | 41 | 33 | **102** | **35.29%** |
| True Negative | 23 | 60 | 83 | **166** | **57.44%** |
| False Positive | 0 | 0 | 5 | **5** | **1.73%** |
| False Negative | 4 | 7 | 5 | **16** | **5.54%** |
| *Statistics* | | | | | |
| Sensitivity | 87.50% | 85.42% | 86.84% | **86.44%** | |
| Specificity | 100.00% | 100.00% | 94.32% | **97.08%** | |
| PPV* | 100.00% | 100.00% | 86.84% | **95.33%** | |
| NPV** | 85.19% | 89.55% | 94.32% | **91.21%** | |

*PPV: positive predictive value.

**NPV: negative predictive value.

## Relationship between uRRR and UTI diagnostic classification

First, assessments were conducted to determine whether there were any demographic differences between participants in the four UTI diagnostic groups. Chi square analysis (**Table 5**) revealed no significant differences between the UTI diagnostic classifications based on sex ($X^2 = 3.218$, df = 3, $p = 0.3593$), age ($X^2 = 27.42$, df = 21, $p = 0.1574$), and reported history of bladder or kidney stones ($X^2 = 0.5756$, df = 3, $p = 0.9020$); however, there were differences based on bladder management ($X^2 = 22.09$, df = 6, $p = 0.0012$), catheter use time ($X^2 = 33.23$, df = 12, $p = 0.0009$), and UTIs per year ($X^2 = 27.91$, df = 9, $p = 0.0010$). When uRRR was assessed across UTI+ ($n = 54$), UTI- ($n = 50$), UTI Indeterminate ($n = 83$), and UTI Unlikely ($n = 102$) groups using a one-way ANOVA, there was a statistically significant difference in uRRR between at least two groups ($F (3, 345) = 38.80$, $p < 0.0001$). Tukey's test for multiple comparisons showed that the mean uRRR values of the UTI+ group differed significantly from those of the UTI-, UTI unlikely, and UTI indeterminate groups ($p < 0.0001$). While

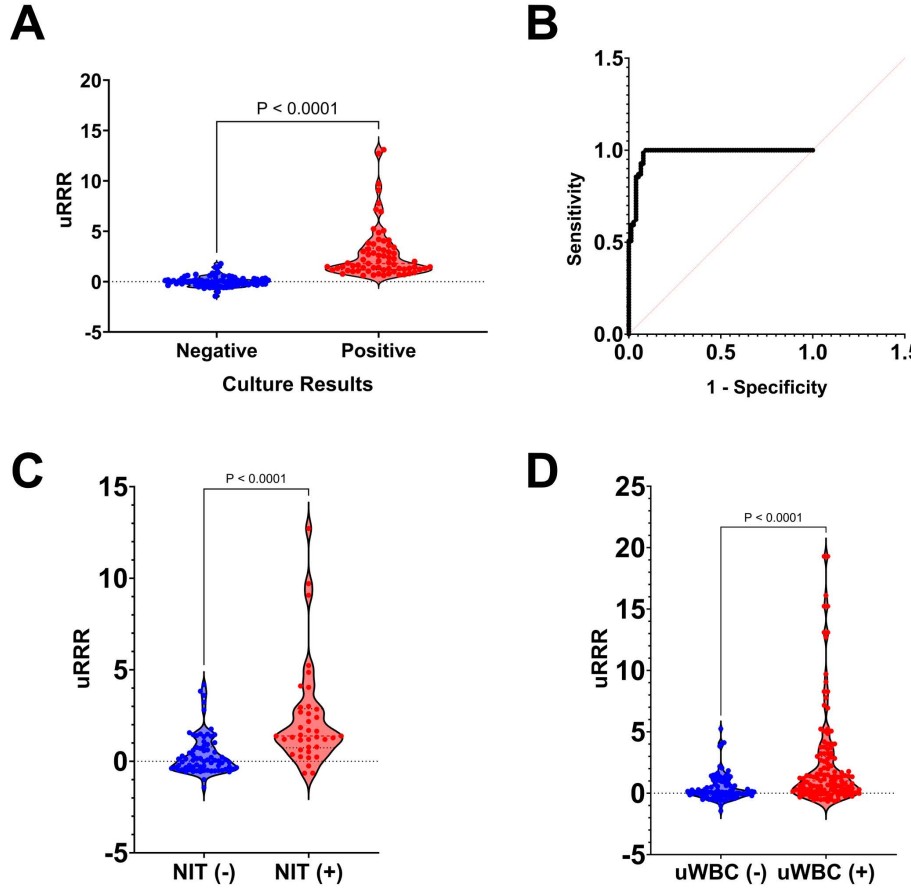

**Fig 2. uRRR versus SUC and UA results. (A)** All patient samples (*n* = 289) were grouped based on urine culture results and uRRR values were compared in a violin plot. **(B)** ROC curve of the TPR against the FPR for uRRR readings versus urine culture results. The AUC of the ROC analysis was 0.9785 (95% CI: 0.9579 to 0.9990, *p* < 0.0001). **(C)** Violin plot comparing uRRR values to UA-based nitrite. **(D)** Violin plot comparing uRRR values to uWBC.

the UTI+ group had the highest uRRR values (3.750, (−0.1500, 19.29), 95% CI: 2.517–4.983), there was a trend toward decreasing uRRR values for: UTI indeterminate (0.9286, (−0.6500, 6.940), 95% CI: 0.6542–1.203), UTI unlikely (0.2506, (−1.440, 15.91), 95% CI: 0.008318–0.4928) and UTI- (0.02215, (−0.5900, 1.510), 95% CI: −0.07388–0.1182) groups, respectively (**Fig 4**).

### Identified species profiles

31% of the samples in this study were polymicrobial, which is consistent with the evidence (up to 39% of UTIs are polymicrobial [18]) (**Table 6**). Further, the relative proportions of identified bacteria in samples differed among bladder management groups and USQNB-based risk profiles (**Fig 5**).

### Discussion

In this first-in-human study among people with NLUTD in both the inpatient and outpatient settings, uRRR is demonstrated to be associated with (1) symptom complexes indicative of higher likelihood of UTI; (2) the presence of uWBC and nitrite

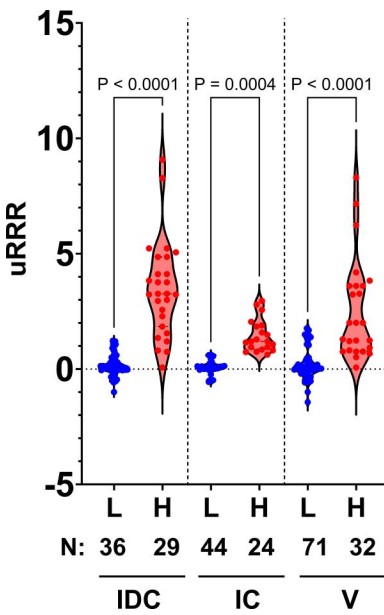

**Fig 3. Analysis of uRRR in samples by bladder management group and USQNB profile.** Violin plot showing differences in uRRR values in each bladder management group (IDC, IC, V). The number of samples comprising each group are indicated under the lower axis. uRRR values were compared using Kolmogorov-Smirnov tests.

in urine detected by urinalysis; and (3) bacterial growth detected by SUC. Additionally, uRRR levels differentiated UTI+ from indeterminate, unlikely and UTI-, and there was a trend toward decreasing uRRR values across all 4 UTI definitions. Despite emerging limitations and TTR of 1–3 days, UA and SUC continue to be the standard of care laboratory tests for UTI, as no other laboratory-based diagnostic solutions have been approved by the FDA and adopted into clinical practice. Thus, currently they remain the appropriate clinical comparators.

First, regarding uRRR as a biomarker for bacterial growth, it was demonstrated that calibration using spiked urine samples to determine the quantification of uRRR revealed a linear and dose-dependent correlation between RFU and CFU cut points for each tested strain. Additionally, the limit of detection for uRRR was determined to be $10^2$ CFU/mL, which is below the clinical threshold typically indicative of infection ($10^5$ CFU/mL). These results confirm that uRRR behaves as designed, is highly reproducible, and precise as within-lab coefficient of variation was below 15% (see Supplemental Table 3 for all %CV values, Supplemental Results: Evaluation of uRRR quantification in S1 File).

An association was observed between uRRR and *a priori* definitions of UTI using a combination of symptoms, uWBC, and bacterial growth (which is not surprising given the relationships between uRRR and each individual diagnostic criterion). Importantly, uRRR differentiated UTI+ from the UTI indeterminate, unlikely, and negative groups. Further, a trend was observed across all four groups, and although there were not statistically significant differences between the UTI indeterminate, unlikely and negative groups, a larger sample size may yield more conclusive results in differentiating these groups. The clinical relevance of these results is the potential for uRRR to discriminate those urine samples that may be representative of UTI versus those that may be prodromal (and potentially amenable to non-antibiotic approaches such as increased fluid intake, etc.) versus non-infectious urinary symptoms. Further, as uRRR has a TTR of one hour, this has the potential to be highly clinically relevant by providing clinically actionable quantitative information at point of care. In contrast, coupled with lengthy TTR, the evidence base indicates that UA and SUC have limited value for complicated UTI diagnosis among people with NLUTD, as both are frequently positive in the absence of clinical symptoms [9,10].

**Table 5. Comparison of UTI diagnostic classifications with patient demographics.**

| Category | UTI (+) (n = 54) | UTI (-) (n = 50) | UTI Indeterminate (n = 83) | UTI Unlikely (n = 102) | χ2 | df | p |
|---|---|---|---|---|---|---|---|
| **Sex (n and %)** | | | | | 3.218 | 3 | 0.3593 |
| *Male* | 36 (66.7%) | 40 (80.0%) | 55 (66.3%) | 71 (69.6%) | | | |
| *Female* | 18 (33.3%) | 10 (20.0%) | 28 (33.7%) | 31 (30.3%) | | | |
| **Age (n and %)** | | | | | 27.42 | 21 | 0.1574 |
| *18–19 years* | 1 (1.9%) | 0 (0.0%) | 3 (3.6%) | 1 (1.0%) | | | |
| *20–29 years* | 6 (11.1%) | 1 (2.0%) | 6 (7.2%) | 5 (4.9%) | | | |
| *30–39 years* | 14 (25.9%) | 3 (6.0%) | 11 (13.3%) | 12 (11.8%) | | | |
| *40–49 years* | 7 (13.0%) | 11 (22.0%) | 13 (15.7%) | 19 (18.6%) | | | |
| *50–59 years* | 9 (16.7%) | 17 (34.0%) | 15 (18.1%) | 26 (25.5%) | | | |
| *60–69 years* | 13 (24.1%) | 13 (26.0%) | 19 (22.9%) | 22 (21.6%) | | | |
| *70–79 years* | 4 (7.4%) | 5 (10.0%) | 13 (15.7%) | 15 (14.7%) | | | |
| *≥ 80 years* | 0 (0.0%) | 0 (0.0%) | 3 (3.6%) | 2 (2.0%) | | | |
| **Bladder Management (n and %)** | | | | | 22.09 | 6 | 0.0012 |
| *IDC* | 23 (42.6%) | 13 (26.0%) | 35 (42.2%) | 28 (27.5%) | | | |
| *IC* | 15 (27.8%) | 10 (20.0%) | 27 (32.5%) | 19 (18.6%) | | | |
| *V* | 16 (29.6%) | 27 (54.0%) | 21 (25.3%) | 55 (53.9%) | | | |
| **Catheter Use Time (n and %)** | | | | | 33.23 | 12 | 0.0009 |
| *0–1 years* | 28 (51.9%) | 42 (84.0%) | 43 (51.8%) | 74 (72.5%) | | | |
| *1–2 years* | 6 (11.1%) | 1 (2.0%) | 6 (7.2%) | 4 (3.9%) | | | |
| *2–5 years* | 12 (22.2%) | 2 (4.0%) | 6 (7.2%) | 8 (7.8%) | | | |
| *5–10 years* | 4 (7.4%) | 0 (0.0%) | 8 (9.6%) | 5 (4.9%) | | | |
| *> 10 years* | 11 (20.4%) | 6 (12.0%) | 20 (24.1%) | 11 (10.8%) | | | |
| **UTIs Per Year (n and %)** | | | | | 27.91 | 9 | 0.0010 |
| *0–1* | 28 (51.9%) | 39 (78.0%) | 45 (54.2%) | 79 (77.5%) | | | |
| *2–3* | 10 (18.5%) | 7 (14.0%) | 23 (27.7%) | 13 (12.7%) | | | |
| *4–5* | 7 (13.0.%) | 2 (4.0%) | 3 (3.6%) | 2 (2.0%) | | | |
| *>5* | 9 (16.7%) | 2 (4.0%) | 12 (14.5%) | 8 (7.8%) | | | |
| **Bladder or Kidney Stones (n and %)** | | | | | 0.576 | 3 | 0.9020 |
| *Yes* | 3 (5.6%) | 2 (4.0%) | 5 (6.0%) | 4 (4.0%) | | | |
| *No* | 51 (94.4%) | 48 (96.0%) | 78 (94.0%) | 98 (96.0%) | | | |

If uRRR is demonstrated to be a reliable biomarker for complicated UTI in future clinical trials, a major advantage would be the ability to make more informed decisions around antibiotic use. In the NLUTD population, because of the high likelihood of infection and the higher likelihood of hospitalization due to UTI than the general population [1–3], empiric antibiotics are frequently prescribed. However, empiric antibiotics are the incorrect choice 25%−51.3% of the time [19] and have had the unwanted consequence of multidrug-resistant infections. This overprescription and exposure to unnecessary antibiotics contributes to antimicrobial resistance and prolongs patient discomfort. The current state of practice represents "harms associated with indiscriminate use of antibiotics (for presumed UTI) associated with lack of diagnostic clarity." [8]

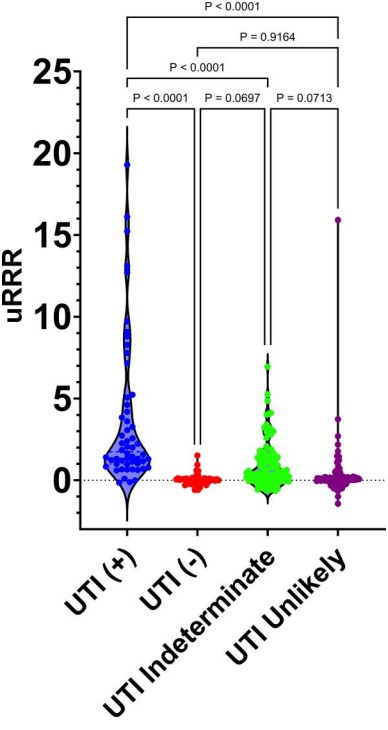

**Fig 4. Analysis of uRRR by UTI diagnostic category.** Violin plots showing differences in uRRR values between UTI diagnostic categories (see Table 2). uRRR values were compared using one-way ANOVA, followed by Tukey's test for multiple comparisons. *P* values for all groups comparisons are indicated.

Highly relevant is that quantification of uRRR represents a measure of bacterial metabolic activity of the *entire urobiome.* In contrast, SUC was developed in the 1950's specifically to identify *E. coli* amongst pregnant women at risk for kidney infections (pyelonephritis), then quickly expanded to urine sampling for bladder infections (cystitis). As the long-held dogma was that "healthy urine is sterile" and an infection represented "invasion of a sterile space by a uropathogen," [20] cultivation of isolated species (particularly *E.coli*) was reasonable. However, contemporary evidence of the urobiome [21–24] demands that clinical testing (and subsequent diagnosis and treatment) consider the entire urine ecosystem, inclusive of bacterial interactions, as opposed to targeted isolation of a narrow selection of bacterial species. Instead of isolating a single bacterial species as done by SUC, uRRR measures the bacterial metabolic activity of the entire urine sample. This takes into account the entirety of the urobiome as well as interactions between bacteria in polymicrobial samples that may alter the activity of bacterial species [20,24,25]. As stated by Brubaker and Wolfe, there is a need to "step away from the simplistic dichotomy that has governed this area of clinical medicine for decades." Further, they state that "a definitive test is needed that is timely (i.e., takes hours) and accurately determines the microbiota present, assessing both relevant commensals and potential uropathogens." [8] While uRRR may achieve some of these goals with the advantage that it assesses the metabolic activities and interactions of commensals and potential uropathogens with the rapid TTR of one hour, identification of the microbiota may require other approaches.

The proportion of polymicrobial samples (31%) (Table 6) underscores the importance of bacterial interactions that can alter responses to antibiotics [25]. Further, the relative proportions of identified bacteria in samples differed among bladder management groups and USQNB-based risk profiles (**Fig 5**), highlighting the importance of considering bladder management and UTI risk stratification when analyzing responses to antibiotics. However, it must be noted that the uRRR

**Table 6. Identified species profile from urine culture by bladder management group.**

| | IDC | | IC | | V | | Total | |
|---|---|---|---|---|---|---|---|---|
| | N | % | N | % | N | % | N | % |
| **Identified** | | | | | | | | |
| *Achromobacter xylosoxidans* | 0 | 0.00% | 0 | 0.00% | 1 | 0.21% | 1 | 0.21% |
| *Acinetobacter spp.* | 1 | 0.21% | 0 | 0.00% | 4 | 0.85% | 5 | 1.07% |
| *Aerococcus sanguinicola* | 0 | 0.00% | 0 | 0.00% | 1 | 0.21% | 1 | 0.21% |
| *Aerococcus urinae* | 0 | 0.00% | 2 | 0.43% | 3 | 0.64% | 5 | 1.07% |
| *Alcaligenes spp.* | 2 | 0.43% | 0 | 0.00% | 1 | 0.21% | 3 | 0.64% |
| *Candida spp.* | 2 | 0.43% | 0 | 0.00% | 1 | 0.21% | 3 | 0.64% |
| *Carynebacterium spp.* | 0 | 0.00% | 0 | 0.00% | 1 | 0.21% | 1 | 0.21% |
| *Citrobacter spp.* | 3 | 0.64% | 1 | 0.21% | 4 | 0.85% | 8 | 1.71% |
| Coagulase Negative *Staphylococcus* | 1 | 0.21% | 3 | 0.64% | 13 | 2.78% | 17 | 3.63% |
| Coliforms | 1 | 0.21% | 0 | 0.00% | 0 | 0.00% | 1 | 0.21% |
| *Corynebacterium spp.* | 2 | 0.43% | 2 | 0.43% | 5 | 1.07% | 9 | 1.92% |
| *Delftia acidovorans* | 0 | 0.00% | 0 | 0.00% | 1 | 0.21% | 1 | 0.21% |
| Diphtheroids | 0 | 0.00% | 2 | 0.43% | 0 | 0.00% | 2 | 0.43% |
| *Enterobacter spp.* | 1 | 0.21% | 1 | 0.21% | 4 | 0.85% | 6 | 1.28% |
| *Enterococcus faecalis* | 20 | 4.27% | 7 | 1.50% | 29 | 6.20% | 56 | 11.97% |
| *Escherichia coli* | 20 | 4.27% | 34 | 7.26% | 25 | 5.34% | 79 | 16.88% |
| *Globicatella spp.* | 0 | 0.00% | 0 | 0.00% | 1 | 0.21% | 1 | 0.21% |
| *Klebsiella pneumoniae* | 9 | 1.92% | 18 | 3.85% | 10 | 2.14% | 37 | 7.91% |
| *Klebsiella spp.* | 3 | 0.64% | 3 | 0.64% | 7 | 1.50% | 13 | 2.78% |
| *Lactobacillus spp.* | 0 | 0.00% | 1 | 0.21% | 4 | 0.85% | 5 | 1.07% |
| *Morganella morganii* | 4 | 0.85% | 0 | 0.00% | 1 | 0.21% | 5 | 1.07% |
| *Proteus mirabilis* | 14 | 2.99% | 3 | 0.64% | 9 | 1.92% | 26 | 5.56% |
| *Proteus spp.* | 0 | 0.00% | 1 | 0.21% | 2 | 0.43% | 3 | 0.64% |
| *Providencia spp.* | 6 | 1.28% | 0 | 0.00% | 0 | 0.00% | 6 | 1.28% |
| *Pseudomonas aeruginosa* | 30 | 6.41% | 3 | 0.64% | 10 | 2.14% | 43 | 9.19% |
| *Pseudomonas spp.* | 1 | 0.21% | 0 | 0.00% | 0 | 0.00% | 1 | 0.21% |
| *Raultella spp.* | 0 | 0.00% | 0 | 0.00% | 1 | 0.21% | 1 | 0.21% |
| *Serratia spp.* | 4 | 0.85% | 0 | 0.00% | 1 | 0.21% | 5 | 1.07% |
| *Staphylococcus aureus* (MRSA) | 2 | 0.43% | 2 | 0.43% | 1 | 0.21% | 5 | 1.07% |
| *Staphylococcus spp.* | 1 | 0.21% | 3 | 0.64% | 7 | 1.50% | 11 | 2.35% |
| *Stenotrophomonas maltophilia* | 1 | 0.21% | 0 | 0.00% | 1 | 0.21% | 2 | 0.43% |
| *Streptococcus spp.* | 3 | 0.64% | 9 | 1.92% | 7 | 1.50% | 19 | 4.06% |
| **Other** | | | | | | | | |
| No Growth | 12 | 2.56% | 31 | 6.62% | 37 | 7.91% | 80 | 17.09% |
| Non-Uropathogenic Gram Positive Organism | 0 | 0.00% | 0 | 0.00% | 4 | 0.85% | 4 | 0.85% |
| Not Performed | 1 | 0.21% | 1 | 0.21% | 1 | 0.21% | 3 | 0.64% |

contains convoluted signals from multiple interactions between bacterial species in mixed cultures. Our analytical characterization of the uRRR has only explored signals derived from pairs of bacterial species; however, it has not yet been deconvoluted for mixed cultures with more than two predominant species. Therefore, the influence of mixed bacterial populations on uRRR measurement and interpretation cannot be fully characterized relative to the bacterial composition of mixed cultures.

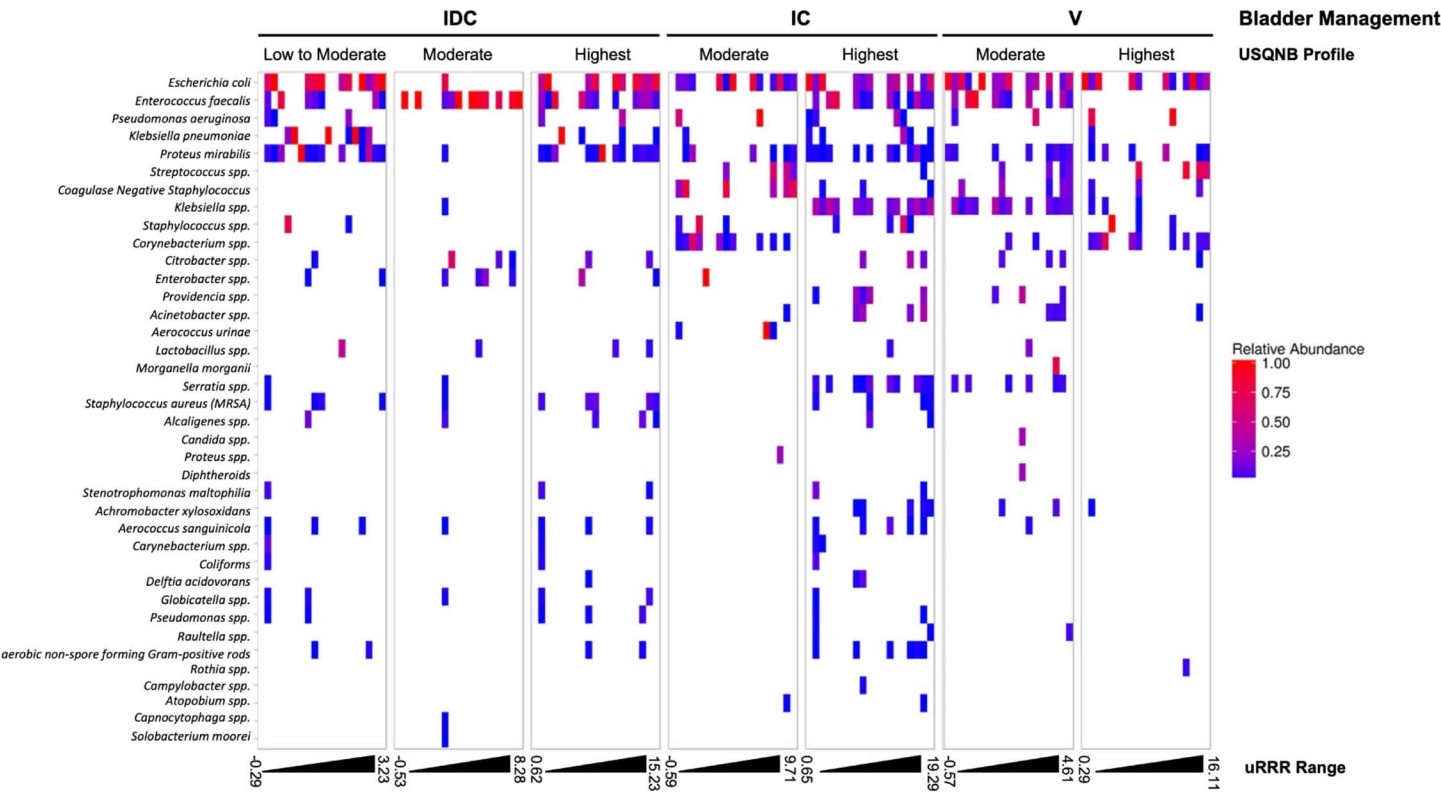

**Fig 5. Relative abundance of species identified in samples by bladder management group and USQNB profile.** Heat map demonstrating the relative abundance of species identified by SUC, stratified by bladder management group, and USQNB profile. The range of uRRR values for each category is represented at the bottom of the heat map, with the range split into 20 bins of equal size. Relative abundances of each identified species (shown on the left axis) are displayed as colors ranging from red to blue as shown in the key.

As noted, a correlation between uRRR and uWBC was demonstrated. The IDSA recommends against using uWBC to differentiate asymptomatic bacteriuria from UTI since inflammation detected as uWBC is frequently elevated in the absence of symptoms among people with NLUTD [4]. However, among people with NLUTD due to spinal cord injury, evidence also suggests that a negative urine dipstick (defined as absence of both leukocyte esterase and nitrite) may have utility in ruling out UTI [26]. Indeed, the clinical situation of the presence of urinary symptoms in the absence of a host inflammatory response is reassuring that an infection is unlikely (and the symptoms may be due to an alternative etiology). Therefore, since the evidence indicates that only an absence of inflammation (as measured by uWBC count), the uWBC cutpoint of ≤5/hpf was chosen to differentiate "positive" from "negative" for the purposes of this work. These preliminary results are encouraging as a promising biomarker for inflammation related to UTI. Other biomarkers under investigation, such as urine neutrophil gelatinase-associated lipocalin (uNGAL), interleukin 8 (IL-8) and interleukin 1β (IL-1β) require further development before readiness for clinical application in adults. While the latter two biomarkers remain experimental, uNGAL was granted 510(k) approval by the FDA in 2023 as a biomarker for diagnosing acute kidney injury in those 3 months to 22 years [27]. While uNGAL is a promising biomarker for acute kidney injury, challenges to its use for UTI diagnosis include its overlap with other conditions (such as sepsis and cancer) and a relatively narrow time range of positive results [28,29]. Recently, it has been proposed that the presence of two or more of the following biomarkers: elevated uNGAL, IL-8, and IL-1β in symptomatic individuals with positive SUC and/or multiplex PCR (M-PCR) is representative of UTI [30]. While this approach deserves further study, relying

on 3 unique biomarkers has the potential for greater complexity, TTR, and cost. Further, molecular rapid diagnostics, such as M-PCR panels, have high sensitivity and specificity, but do not provide host inflammatory response information, and have the disadvantages of high cost and potential overdiagnosis due to detection of DNA from both live and dead bacteria [20].

Limitations of this study include its cross-sectional design, single-center setting, the representativeness of the study population, and, in the case of people with indwelling catheters, that urine sampling was directly from the urine collection bag. The cross-sectional study design limits the conclusions that can be drawn from this data in terms of causality. In addition, data was collected from a single site, which can limit the representativeness of the study population. As stated in the methods, urine sampling directly from the urine collection bag in IDC was intentional as the study was designed to assess the performance of the assay under conditions in which there was high variability of hypothesized bacterial metabolic activity (it was suspected that urine samples obtained from the collection bag of individuals with indwelling catheters would be biomass-heavy compared with those of non-catheter users). This is not clinical standard of care, so any results relevant to this population should not be generalized to patient-level diagnostic recommendations without further validation, using clinically collected samples. As this is a preliminary study of the agreement between standard clinical assessments and uRRR, sampling urine from the urine collection bag should not change the agreement between these two measures. Furthermore, although this study utilized regression modeling with backward selection to identify associations between variables, an alternative modeling approach, such as using Directed Acyclic Graphs (DAGs) may be more appropriate with larger studies that explore relationships between a large number of explanatory variables. The use of this visual tool would allow full consideration of relationships between exposure, outcomes, confounders, bias, and causation. These factors are not completely considered in this analysis, although propensity score methods are used to minimize the impact of potential confounders. Propensity scoring methods also only considered bladder management method and catheter use duration as potential confounders; however, multivariable analysis on bladder management method, duration of catheterization, prior antibiotic use, and comorbidities were not performed, which may affect urine microbiota and inflammatory markers. A strength of this study is the focus on the NLUTD population, who are at high risk of complicated (and often, catheter-associated) UTI, experience recurrent UTI, and often are more challenging to diagnose (and treat). Further research is needed on whether RFU cut points differ between bladder management groups and in different patient populations.

## Conclusions

This is the first report of uRRR as a potential rapid diagnostic assessment tool for urobiome dysbiosis consistent with UTI among people with NLUTD, a population with high burden and unmet need due to complicated UTI. Preliminary relationships between uRRR and urine uWBC, urine nitrite, bacterial growth (in CFU/mL), symptoms and likelihood of UTI were demonstrated here. URRR's advantages include rapid TTR and culture-independent metabolic signal. Given these promising results, further exploration is warranted to determine the contributions of uRRR to clinical decision-making around UTI among people with NLUTD.

## Supporting information

**S1 File. Supplemental Methods and Results.** Additional pre-clinical materials discussing the methods used to establish the experimental assay and assess assay linearity and precision.
(DOCX)

**S1 Fig. Distribution of propensity scores for UTI risk groups.** Density plot of propensity scores based on UTI risk groups, constructed as a visual inspection of balance.
(TIF)

## Author contributions

**Conceptualization:** Abigail P. Fox, Ana Valeria Aguirre Guemez, Courtney Cavin, Suzanne L. Groah.

**Data curation:** Abigail P. Fox, Ana Valeria Aguirre Guemez, Christopher Skipwith, Courtney Cavin, Suzanne L. Groah.

**Funding acquisition:** Courtney Cavin, Suzanne L. Groah.

**Investigation:** Abigail P. Fox, Ana Valeria Aguirre Guemez, Christopher Skipwith, Courtney Cavin, Suzanne L. Groah.

**Methodology:** Abigail P. Fox, Ana Valeria Aguirre Guemez, Suzanne L. Groah.

**Project administration:** Ana Valeria Aguirre Guemez, Suzanne L. Groah.

**Resources:** Abigail P. Fox, Ana Valeria Aguirre Guemez, Christopher Skipwith, Courtney Cavin, Suzanne L. Groah.

**Software:** Christopher Skipwith, Courtney Cavin.

**Supervision:** Ana Valeria Aguirre Guemez, Christopher Skipwith, Suzanne L. Groah.

**Validation:** Abigail P. Fox, Ana Valeria Aguirre Guemez, Courtney Cavin, Suzanne L. Groah.

**Visualization:** Abigail P. Fox, Christopher Skipwith.

**Writing – original draft:** Abigail P. Fox, Ana Valeria Aguirre Guemez, Christopher Skipwith, Courtney Cavin, Suzanne L. Groah.

**Writing – review & editing:** Abigail P. Fox, Ana Valeria Aguirre Guemez, Christopher Skipwith, Courtney Cavin, Suzanne L. Groah.

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
