## [Decision Letter · Decision Letter 0]

12 Sep 2025

Dear Dr. Fox,

Thank you for submitting your manuscript to PLOS ONE. After careful consideration, we feel that it has merit but does not fully meet PLOS ONE’s publication criteria as it currently stands. Therefore, we invite you to submit a revised version of the manuscript that addresses the points raised during the review process.

We look forward to receiving your revised manuscript.

Kind regards,

Eva Torres-Sangiao, PhD

Academic Editor

PLOS ONE

“This study was funded by Astek Diagnostics, Baltimore, MD.”

“This work was funded by Astekdx, LLC (https://astekdx.com). Astekdx did not play a role in study design or data collection. Astekdx was involved in the analyses, decision to publish and preparation of the manuscript.”

“I have read the journal's policy and the authors of this manuscript have the following competing interests: CS serves as CSO and receives salary from Astekdx; CC serves as COO and receives salary from Astekdx; SLG serves as CMS and has future equity shares in Astekdx.”

Reviewers' comments:

Reviewer's Responses to Questions

**Comments to the Author**

1. Is the manuscript technically sound, and do the data support the conclusions?

Reviewer #1: Partly

Reviewer #2: Yes

Reviewer #3: Partly

Reviewer #4: Yes

Reviewer #5: Yes

2. Has the statistical analysis been performed appropriately and rigorously?

Reviewer #1: Yes

Reviewer #2: Yes

Reviewer #3: N/A

Reviewer #4: Yes

Reviewer #5: Yes

3. Have the authors made all data underlying the findings in their manuscript fully available?

Reviewer #1: Yes

Reviewer #2: Yes

Reviewer #3: No

Reviewer #4: Yes

Reviewer #5: Yes

4. Is the manuscript presented in an intelligible fashion and written in standard English?

Reviewer #1: Yes

Reviewer #2: Yes

Reviewer #3: Yes

Reviewer #4: Yes

Reviewer #5: Yes

Reviewer #1: To Author,

This study explores the potential of the urine resazurin reduction ratio (uRRR) as a rapid diagnostic tool for urinary tract infec-tions (UTI) in individuals with neurogenic lower urinary tract dysfunction (NLUTD). While the research addresses a significant clinical need and presents an innovative ap-proach, the manuscript, even with the provided supplementary materials, contains sev-eral critical flaws and areas requiring substantial improvement. Therefore, I recom-mend a Major Revision.

Major Comments:

1. Fundamental Flaw in Urine Sample Collection for Indwelling Catheter Patients (Lines 123-124, 343-347):

The collection of urine samples directly from collection bags for individuals with in-dwelling catheters (IDC) constitutes a fatal methodological flaw. This practice is not clinically acceptable due to the high risk of bacterial overgrowth and environmental contamination within the collection bag. Such contamination can lead to false-positive standard urine culture (SUC) results, which serve as the "gold standard" comparator in this study, thereby misrepresenting the true bladder microbiome.The authors' justifica-tion that this approach assesses performance under variable bacterial metabolic activity or that it should not alter agreement between measures is insufficient. The reliability of SUC, and consequently the validity of uRRR's accuracy, sensitivity, and specificity calculations, is severely compromised.

Recommendation: This issue is paramount. The authors must either:

Recollect samples: Ideally, sterile urine samples from IDC patients should be obtained using clinically accepted aseptic techniques (e.g., from a freshly placed catheter or a disinfected catheter port). This would likely necessitate a new study.

Severely qualify conclusions: If recollection is not feasible, the authors must explicitly and prominently acknowledge this fundamental limitation in the abstract, results, and discussion sections. All conclusions pertaining to the IDC patient group must be sig-nificantly qualified, acknowledging that SUC results from collection bags may not accurately reflect true UTI status, thereby limiting the generalizability and clinical utility of uRRR in this specific context.

2. Illegible Figures and Lack of High-Resolution Presentation (All Figures):

All figures (Figures 1, 2, 3, 4, 5) are presented in extremely low resolution, rendering them blurry and pixelated. Critical details, including axis labels, data points, statistical annotations (e.g., p-values on violin plots), and legends, are completely indistin-guishable. For example, in Figure 5 (heatmap), the "uRRR Range" values at the bottom and the specific species names on the left axis are unreadable.

Recommendation: The authors must provide high-resolution versions of all figures. All textual elements, data points, and graphical components within the figures must be sharp, clear, and easily readable. This is a non-negotiable prerequisite for any further consideration of the manuscript.

3. Insufficient Transparency Regarding Funding and Competing Interests (Page 2, 3, 4, 29):

The study was funded by Astekdx, LLC, which also participated in data analysis, publication decisions, and manuscript preparation. Furthermore, authors CS, CC, and SLG have direct financial interests with Astekdx. While declared in the submission metadata, the potential for bias when a company is so deeply involved in research on its own diagnostic tool is substantial.

Recommendation: The authors must include a dedicated paragraph in the "Discussion" section of the main manuscript explicitly acknowledging the funding source and the authors' financial relationships with Astekdx. This discussion should address how these relationships might influence the interpretation of the study's findings and underscore the critical need for independent validation of these results in future research.

4. Methodological Rigor: Confounding Factors and Multiple Imputation:

Transparency in Confounding Assessment: The supplementary materials mention the use of propensity score methods and instrumental variables to address confounding. While this is a positive step, the manuscript lacks transparency in how potential con-founders were identified and handled.

Recommendation: The authors should consider using Directed Acyclic Graphs (DAGs) to visually represent their assumptions about causal relationships and potential con-founding pathways. This would enhance transparency in their analytical approach to confounding and provide a clear rationale for the variables included in their statistical models. This explanation, along with the specific methods used (propensity scores, instrumental variables), should be clearly detailed in the main "Methods" section.

Detailed Reporting of Multiple Imputation: The supplementary materials indicate the use of multiple imputation for missing data. However, the specific operational details of this procedure are absent.

Recommendation: The "Methods" section must include explicit details of the multiple imputation procedure. This should encompass: (a) the imputation model used (e.g., variables included, type of imputation algorithm), (b) the number of imputations per-formed, and (c) any diagnostics used to assess convergence.

Impact of Imputation on Conclusions: It is crucial to assess whether the imputation process significantly influenced the study's conclusions.

Recommendation: The authors should conduct and report sensitivity analyses com-paring the results obtained from the imputed dataset with those from a complete-case analysis (if feasible and appropriate). This would provide valuable insight into the ro-bustness of their findings and whether the imputation procedure altered the study's primary conclusions.

5. Integration of uRRR Clinical Cut-Point Determination (Lines 132-135):

The supplementary materials provide a detailed description of how the uRRR clinical cut-points were determined, including the use of single-species clinical samples, a 3x3 contingency table, weighted Cohen's κ statistic, and the "Index of Union (IU)" method.

Recommendation: While this information is now available, such critical methodological details must be clearly summarized and described within the "Methods" section of the main manuscript. Readers should be able to understand the derivation of these cut-points from the main text, with detailed calculations and raw data retained in the supplementary materials. The specific RFU values (0.7310 and 0.5469) and their clin-ical interpretation should be explicitly stated in the main manuscript.

6. Inconsistency Between Study Design Limitations and Strong Diagnostic Claims:

The manuscript presents strong diagnostic performance metrics (e.g., overall accuracy, sensitivity, specificity) while simultaneously stating that it is a "cross-sectional study" and a "first-in-human study" aiming to demonstrate "preliminary relationships." This juxtaposition can lead to an overinterpretation of uRRR's diagnostic utility.

Recommendation: The phrasing in the abstract, results, and discussion should be care-fully revised to align diagnostic utility claims with the cross-sectional study design. While the observed associations are encouraging, direct claims of "diagnosis" or "rapid diagnostic assessment tool" should be more cautiously phrased as "potential" or "promising for future development." The discussion (Lines 355-361) should further elaborate on this distinction.

7. Clarity on Research Necessity, Innovation, and Contribution to the Field (Intro-duction & Discussion):

While the unmet need in NLUTD is mentioned, the Introduction could more robustly articulate the precise necessity and innovation of uRRR beyond simply being a rapid test. Similarly, the Discussion needs to more clearly delineate how these findings ad-vance the field compared to existing literature.

Recommendation:

Introduction: Enhance the introduction to specifically detail why existing diagnostic methods are particularly problematic for NLUTD patients (e.g., altered symptom presentation, asymptomatic bacteriuria, polymicrobial infections, limitations of current diagnostics in this specific population). Clearly articulate how uRRR's mechanism (measuring the metabolic activity of the entire urobiome) offers a distinct advantage over traditional SUC (which often focuses on single isolates) or dipstick tests, thereby highlighting its specific innovation.

Discussion: Expand the discussion to explicitly compare the novel findings with ex-isting literature, particularly regarding the challenges posed by polymicrobial infections and the evolving "urobiome" concept (Lines 273-288, 328-333). Emphasize how uRRR's ability to assess overall metabolic activity introduces a new dimension to un-derstanding UTI in NLUTD, potentially offering deeper insights than mere CFU counts or single biomarkers. Discuss how these preliminary findings lay the groundwork for a potential paradigm shift in NLUTD UTI diagnosis.

Minor Comments:

1. Evidence for "Reproducibility and Precision" Claim (Line 272):

The supplementary materials provide detailed precision (%CV) data (Table 3), including within-day, between-operator, and between-instrument variability, and state that "within-lab variation below 15%."

Recommendation: Since supporting data are available, the authors should explicitly refer to these precision data (e.g., stating the overall %CV was below 15%) in the "Results" section of the main manuscript and cite the detailed tables in the supplemen-tary materials.

2. Clarity of "Limit of Detection" Statement (Line 270):

The phrasing "Additionally, the limit of detection (10^2 CFU/mL) was below the threshold typically used as indicative of infection (10^5 CFU/mL)" is slightly awkward.

Recommendation: Rephrase for clarity, e.g., "The limit of detection for uRRR was de-termined to be 10^2 CFU/mL, which is below the conventional clinical threshold of 10^5 CFU/mL typically used to indicate infection, suggesting high analytical sensitiv-ity."

3. Mislabeling of Figure 2 Plots:

The term "Volcano plot" is incorrectly used in the figure legends and text (Lines 211, 214, 215, 229, 254) for Figure 2 (A, C, D). These are clearly violin plots.

Recommendation: Change "Volcano plot" to "Violin plot" in all relevant figure legends and text references.

4. Grammar and Spelling Errors:

Line 52: "except when either value are negative." should be corrected to "either value is negative."

Line 76: "Filter-sterilized-filtered urine" is redundant. It should be revised to "Fil-ter-sterilized urine."

Line 127: "≥6 hpfuWBC" requires a space for clarity, e.g., "≥6 hpf uWBC."

Line 188: "lr with_95% confidence interval" should be rephrased for standard academic style, e.g., "linear regression with 95% confidence interval."

Line 247: "While the UTI+" appears to be an incomplete sentence. Although it connects to Line 248, the paragraph break is grammatically awkward.

5. Table 5 (Patient demographics) Percentage Representation:

The "n and row %" presentation in Table 5 for "Sex, Age, Bladder Management, Catheter Use Time, and UTIs Per Year" is ambiguous regarding the base of the per-centage.

Recommendation: It should be clarified whether the percentage refers to the total number of individuals in that demographic category or the percentage within each UTI diagnostic group. Adopting standard reporting practices, such as presenting N and column percentages, would enhance clarity.

Reviewer #2: The manuscript is overall well-written and addresses an important clinical problem. However, several methodological and results-related issues should be addressed before acceptance:

1. The cross-sectional design limits the ability to infer causality. Some statements in the abstract and discussion suggest causal interpretation, which should be revised to avoid overstatement.

2. Potential confounding factors such as bladder management method, duration of catheterization, prior antibiotic use, and comorbidities may affect urine microbiota and inflammatory markers. These factors were only descriptively presented, without adequate statistical adjustment. Either additional multivariable analysis should be performed or the limitation clearly acknowledged.

3. The determination of uRRR cut-off values requires more detail. The rationale for the chosen thresholds, validation across bacterial species, and reproducibility testing (intra- and inter-assay variability) are not sufficiently explained.

4. Approximately one-third of the samples were polymicrobial, but the influence of mixed bacterial populations on uRRR measurement and interpretation was not discussed. This aspect should be clarified, as it is clinically relevant in NLUTD patients.

5. The results section sometimes emphasizes statistical significance without presenting effect sizes or clinical relevance. Greater attention should be given to absolute differences, confidence intervals, or effect magnitudes to improve interpretation.

6. The discussion should expand on how uRRR could be integrated into existing diagnostic workflows, including considerations of cost, feasibility, and potential barriers in clinical practice.

7. The limitations section should be expanded. In addition to the cross-sectional design, the single-center setting, the representativeness of the study population, and the possible bias from urine collection via indwelling catheter bags should all be explicitly mentioned.

8. The abstract could be improved by highlighting the novelty of uRRR as a rapid diagnostic biomarker, emphasizing its clinical implications, and briefly summarizing the main advantages compared with current diagnostic standards, rather than focusing mainly on detailed statistical results.

Reviewer #3: This manuscript addresses an important and clinical relevant problem. The study is well-conceived and the writing is overall clear. The main concern at this stage is that the data are not visible in the file (figures appear cropped).

Since the data represent the core of the manuscript, I recommend a major revision to address this issue, otherwise, the manuscript cannot be properly evaluated and would need to be rejected. The association between urine resazurin reduction ratio (uRRR), urinary biomarkers, and UTI likelihood is novel and potentially impactful.

Secondary suggestions for improvement:

a) some sentences in the Introduction section are long or contain punctuation issues that affect readability (e.g., line 41). Please carefully revise the language and avoid repetition (Introduction and Discussion).

b) Check that all the abbreviations are introduced at the first mention and then used consistently throughout the text.

c)The Results section is highly technical. However, since the figures are not visible in the current submission, I cannot fully evaluate it. This should be corrected before the next revision.

Reviewer #4: In the clinical diagnosis of urinary tract infections, commonly used indicators include urinary white blood cells, nitrites, and urine culture. The content of this study is novel, providing a new tool for the rapid diagnosis of urinary tract infections in patients with neurogenic bladder. Of course, further exploration is needed for its clinical application.

Reviewer #5: Title

Urine resazurin reduction ratio is associated with urine biomarkers and likelihood of urinary tract infection in individuals with neurogenic bladder - not convincing title and clear title. The title doesnt explain the work.

Introduction

"For people with neurogenic lower urinary tract dysfunction (NLUTD). UTI is" not an apt sentence to start, and the sentence is not continuous and is incomplete.

"(SUC)),(4)"—double bracket with reference quoted outside period.

Introduction doesn't speak about urine biomarkers taken into study. What and when Urine resazurin reduction ratio is used. The introduction should include topics relevant to the title.

Methodolgy

Methodology is unclear. Methodology should list the study population, number of samples, inclusion and exclusion crietria, materials used, Number of urine samples obtained. The methods be listed out first.

Filter-sterilized urine samples - references on the method used obtain the sample.

Reason out the experimental and clinical methods taken in the study.

Discussion should discuss the outcome of this study compared with any previous studies. Any discussion other than said should be removed. First few paragraphs are more an introduction rather than discussion. Discussion should be discussing the uRRR carried out with any other studeis. Table in discussion should be to compare the results with other studies.

In the manuscript, the word WE should not be used.

**Do you want your identity to be public for this peer review?** For information about this choice, including consent withdrawal, please see our Privacy Policy

Reviewer #1: No

Reviewer #2: **Yes:** 玄汉胡

Reviewer #3: No

Reviewer #4: No

Reviewer #5: No

---

## [Author Response · Author response to Decision Letter 1]

3 Nov 2025

Thank you for your thoughtful review of our article entitled “Urine Resazurin Reduction Ratio as a Biomarker of Urinary Tract Infection in People with Neurogenic Bladder" (manuscript# PONE-D-25-29880). Your suggestions have improved the quality and clarity of our presentation of the data. See below your comments and our modifications. A track changes version and a clean version have been included in the resubmission for your review.

We noted the issues the reviewers had viewing the figures and apologize for the inconvenience. We have trialed various resolutions, file types, and file sizes within PLOS One ranges. We have observed that if the figure is clicked on and downloaded from the compiled PDF, it is clear. However, if viewed within the built pdf document, they appear to have reduced resolution. We are advising reviewers to please click on the figure for best viewing. If accepted, we will work with the copy editor staff to resolve this issue.

Reviewer 1

Fundamental Flaw in Urine Sample Collection for Indwelling Catheter Patients (Lines 123-124, 343-347):

The collection of urine samples directly from collection bags for individuals with in-dwelling catheters (IDC) constitutes a fatal methodological flaw. This practice is not clinically acceptable due to the high risk of bacterial overgrowth and environmental contamination within the collection bag. Such contamination can lead to false-positive standard urine culture (SUC) results, which serve as the "gold standard" comparator in this study, thereby misrepresenting the true bladder microbiome. The authors' justification that this approach assesses performance under variable bacterial metabolic activity or that it should not alter agreement between measures is insufficient. The reliability of SUC, and consequently the validity of uRRR's accuracy, sensitivity, and specificity calculations, is severely compromised.

Recommendation: This issue is paramount. The authors must either:

Recollect samples: Ideally, sterile urine samples from IDC patients should be obtained using clinically accepted aseptic techniques (e.g., from a freshly placed catheter or a disinfected catheter port). This would likely necessitate a new study.

Severely qualify conclusions: If recollection is not feasible, the authors must explicitly and prominently acknowledge this fundamental limitation in the abstract, results, and discussion sections. All conclusions pertaining to the IDC patient group must be significantly qualified, acknowledging that SUC results from collection bags may not accurately reflect true UTI status, thereby limiting the generalizability and clinical utility of uRRR in this specific context.

Response: We appreciate this important concern and would like to clarify the rationale for our urine sampling strategy. We fully agree that urine collection from the drainage bag is not the recommended standard of care practice for patient diagnosis in the clinical setting, where minimizing contamination is essential. However, our study was designed with a different and specific methodological goal: to evaluate the performance of a new diagnostic device under conditions where polymicrobial urine specimens—often deemed “contaminants” by routine clinical microbiology laboratories—were intentionally included. While the reviewer correctly points out that “contamination can lead to false-positive SUC results”, it is important to note that clinical laboratories will typically assume that polymicrobial growth on SUC represents contamination, when the reality is in this population of people with NLUTD who use indwelling catheters, polymicrobial UTIs are commonplace and do not necessarily represent contamination. It is therefore important to include polymicrobial samples as these are a real-world clinical issue among this population.

Hence, there were multiple reasons for this approach:

1. Device performance in real-world complexity. As noted above, patients with NLUTD and chronic indwelling catheters frequently develop polymicrobial bacteriuria and polymicrobial UTI. Clinical laboratories often treat these samples as contaminated (and do not proceed with SUC), yet they represent a common and clinically relevant challenge. We understand this clinical challenge and by sampling from the collection bag, we ensured a sufficient number of polymicrobial specimens to rigorously test whether our device could perform reliably in these scenarios. We recognize that SUC performance may be compromised in this setting, however it is the clinical gold standard that must be tested against for this real problem.

2. Study aim vs. clinical care. Our objective in this first-in-human trial was not to diagnose urinary tract infections in patients, but rather to challenge the device against the broadest possible spectrum of microbial conditions. We therefore intentionally deviated from clinical collection standards in order to meet the experimental aims.

3. Transparency and reproducibility. We clearly described our collection methods in the manuscript so that readers could interpret results with appropriate context. Further, we stratified by bladder management method (and hence, urine collection method), allowing the reader to interpret findings not only by bladder management method, but by urine collection approach. We agree that these findings should not be generalized to patient-level diagnostic recommendations without further validation using clinically collected samples.

In summary, although collection from the drainage bag is not appropriate for clinical diagnosis, this was an a priori decision that we believe was scientifically justified and necessary in this cross-sectional study to rigorously test device performance in polymicrobial specimens. To avoid any potential misinterpretation, we have revised the manuscript to clarify this rationale and to explicitly note the limitations of this approach: in the methods (line numbers 443-448) we have added "for this first-in-human cross-sectional study, we intentionally deviated from clinical standards and collected urine from the collection bag of those participants who use indwelling catheters. This was done to ensure a wide spectrum of microbial conditions and a sufficient number of polymicrobial samples to rigorously assess device performance under these conditions. This study was intended as an initial assessment of correlations (and not diagnosing UTI) to determine appropriateness for a next-stage human trial. " Further, we have modified our statements in the limitations section (line numbers 1560-1562) to the following: “This is not clinical standard of care, so any results relevant to this population should not be generalized to patient-level diagnostic recommendations without further validation, using clinically collected samples.”

Illegible Figures and Lack of High-Resolution Presentation (All Figures):

All figures (Figures 1, 2, 3, 4, 5) are presented in extremely low resolution, rendering them blurry and pixelated. Critical details, including axis labels, data points, statistical annotations (e.g., p-values on violin plots), and legends, are completely indistin-guishable. For example, in Figure 5 (heatmap), the "uRRR Range" values at the bottom and the specific species names on the left axis are unreadable.

Recommendation: The authors must provide high-resolution versions of all figures. All textual elements, data points, and graphical components within the figures must be sharp, clear, and easily readable. This is a non-negotiable prerequisite for any further consideration of the manuscript.

Response: We have reuploaded the figures at a higher resolution. We apologize for the inconvenience during your first review.

Insufficient Transparency Regarding Funding and Competing Interests (Page 2, 3, 4, 29):

The study was funded by Astekdx, LLC, which also participated in data analysis, publication decisions, and manuscript preparation. Furthermore, authors CS, CC, and SLG have direct financial interests with Astekdx. While declared in the submission metadata, the potential for bias when a company is so deeply involved in research on its own diagnostic tool is substantial.

Recommendation: The authors must include a dedicated paragraph in the "Discussion" section of the main manuscript explicitly acknowledging the funding source and the authors' financial relationships with Astekdx. This discussion should address how these relationships might influence the interpretation of the study's findings and underscore the critical need for independent validation of these results in future research.

Response: Thank you for your concern regarding funding. We were given instructions by Plosone to remove any funding-related information from the main manuscript: “Funding information should not appear in the Acknowledgments section or other areas of your manuscript. We will only publish funding information present in the Funding Statement section of the online submission form. Please remove any funding-related text from the manuscript.”

Methodological Rigor: Confounding Factors and Multiple Imputation:

Transparency in Confounding Assessment: The supplementary materials mention the use of propensity score methods and instrumental variables to address confounding. While this is a positive step, the manuscript lacks transparency in how potential con-founders were identified and handled.

Recommendation: The authors should consider using Directed Acyclic Graphs (DAGs) to visually represent their assumptions about causal relationships and potential con-founding pathways. This would enhance transparency in their analytical approach to confounding and provide a clear rationale for the variables included in their statistical models. This explanation, along with the specific methods used (propensity scores, instrumental variables), should be clearly detailed in the main "Methods" section.

Detailed Reporting of Multiple Imputation: The supplementary materials indicate the use of multiple imputation for missing data. However, the specific operational details of this procedure are absent.

Recommendation: The "Methods" section must include explicit details of the multiple imputation procedure. This should encompass: (a) the imputation model used (e.g., variables included, type of imputation algorithm), (b) the number of imputations per-formed, and (c) any diagnostics used to assess convergence.

Impact of Imputation on Conclusions: It is crucial to assess whether the imputation process significantly influenced the study's conclusions.

Recommendation: The authors should conduct and report sensitivity analyses com-paring the results obtained from the imputed dataset with those from a complete-case analysis (if feasible and appropriate). This would provide valuable insight into the robustness of their findings and whether the imputation procedure altered the study's primary conclusions.

Response: Thank you for your concerns. While we do not utilize Directed Acyclic Graphs in our analysis to analyze all potential confounding pathways and causal relationships, we have noted this study limitation in the discussion of the main manuscript, noting that larger observational studies assessing the impacts of a large number of variables would benefit greatly from the approach. Additionally, we have added detailed descriptions of propensity scoring methods to address confounders, in addition to detailed multiple imputation procedures, including the imputation model used, number of imputations performed, and diagnostics to assess convergence. We have also shown the distribution of complete and imputed results in the Supplemental Information.

Integration of uRRR Clinical Cut-Point Determination (Lines 132-135):

The supplementary materials provide a detailed description of how the uRRR clinical cut-points were determined, including the use of single-species clinical samples, a 3x3 contingency table, weighted Cohen's κ statistic, and the "Index of Union (IU)" method.

Recommendation: While this information is now available, such critical methodological details must be clearly summarized and described within the "Methods" section of the main manuscript. Readers should be able to understand the derivation of these cut-points from the main text, with detailed calculations and raw data retained in the supplementary materials. The specific RFU values (0.7310 and 0.5469) and their clinical interpretation should be explicitly stated in the main manuscript.

Response: Thank you for your comment to include the critical methodological details in the main manuscript. We agree that it is important to include these details in the main manuscript and have added a summary of the supplementary materials in the “Methods” section of the main manuscript (lines 341-359, 402-412). We have also added additional references to the Supplemental Methods to make this information more clear.

Additionally, we moved the paragraph describing the uRRR clinical cut-point determination from the supplemental results to the main results, as we agree that this information should be accessible in the main manuscript (lines 559-593) The uRRR cut-points belong in the Results as opposed to the Methods, because these values are determined experimentally.

Lastly, we have added subheaders in our references to the Supplemental Materials for clarity.

Inconsistency Between Study Design Limitations and Strong Diagnostic Claims:

The manuscript presents strong diagnostic performance metrics (e.g., overall accuracy, sensitivity, specificity) while simultaneously stating that it is a "cross-sectional study" and a "first-in-human study" aiming to demonstrate "preliminary relationships." This juxtaposition can lead to an overinterpretation of uRRR's diagnostic utility.

Recommendation: The phrasing in the abstract, results, and discussion should be care-fully revised to align diagnostic utility claims with the cross-sectional study design. While the observed associations are encouraging, direct claims of "diagnosis" or "rapid diagnostic assessment tool" should be more cautiously phrased as "potential" or "promising for future development." The discussion (Lines 355-361) should further elaborate on this distinction.

Response: You are correct that this cross-sectional study cannot make diagnostic claims. We rephrased the sentences mentioning diagnostics to include the words “potential” or “promising”. We also emphasized that we are not making diagnostic claims and our next step will be to perform a prospective trial to inform treatment and diagnostics.

Line 38, 393, 1334, 1584: added “…first report of uRRR as a potential rapid diagnostic assessment tool…”

We have also added a reference to this in the limitations section of the paper (Lines 1560-1562).

Clarity on Research Necessity, Innovation, and Contribution to the Field (Intro-duction & Discussion):

While the unmet need in NLUTD is mentioned, the Introduction could more robustly articulate the precise necessity and innovation of uRRR beyond simply being a rapid test. Similarly, the Discussion needs to more clearly delineate how these findings ad-vance the field compared to existing literature.

Recommendation:

Introduction: Enhance the introduction to specifically detail why existing diagnostic methods are particularly problematic for NLUTD patients (e.g., altered symptom presentation, asymptomatic bacteriuria, polymicrobial infections, limitations of current diagnostics in this specific population). Clearly articulate how uRRR's mechanism (measuring the metabolic activity of the entire urobiome) offers a distinct advantage over traditional SUC (which often focuses on single isolates) or dipstick tests, thereby highlighting its specific innovation.

Discussion: Expand the discussion to explicitly compare the novel findings with ex-isting literature, particularly regarding the challenges posed by polymicrobial infections and the evolving "urobiome" concept (Lines 273-288, 328-333). Emphasize how uRRR's ability to assess overall metabolic activity introduces a new dimension to understanding UTI in NLUTD, potentially offering deeper insights than mere CFU counts or single biomarkers. Discuss how these preliminary findings lay the groundwork for a potential paradigm shift

---

## [Decision Letter · Decision Letter 1]

11 Jan 2026

Dear Dr. Abigail P Fox,

We’re pleased to inform you that your manuscript has been judged scientifically suitable for publication and will be formally accepted for publication once it meets all outstanding technical requirements.

Kind regards,

Tombari Pius Monsi, Ph.D

Academic Editor

PLOS One

**Comments to the Author**

Reviewer #5: All comments have been addressed

2. Is the manuscript technically sound, and do the data support the conclusions?

Reviewer #5: Yes

3. Has the statistical analysis been performed appropriately and rigorously?

Reviewer #5: Yes

4. Have the authors made all data underlying the findings in their manuscript fully available?

Reviewer #5: Yes

5. Is the manuscript presented in an intelligible fashion and written in standard English?

Reviewer #5: Yes

Reviewer #5: (No Response)

**Do you want your identity to be public for this peer review?** For information about this choice, including consent withdrawal, please see our Privacy Policy

Reviewer #5: No

---

## [Editor Report · Acceptance letter]

PONE-D-25-29880R1

PLOS One

Dear Dr. Fox,

I'm pleased to inform you that your manuscript has been deemed suitable for publication in PLOS One. Congratulations! Your manuscript is now being handed over to our production team.

Kind regards,

on behalf of

Dr. Tombari Pius Monsi

Academic Editor

PLOS One